# Impact of deltamethrin-resistance in *Aedes albopictus* on its fitness cost and vector competence

**Jielin Deng**[1☯], **Yijia Guo**[1☯], **Xinghua Su**[1], **Shuang Liu**[1], **Wenqiang Yang**[1], **Yang Wu**[1], **Kun Wu**[1], **Guiyun Yan**[2], **Xiao-Guang Chen**[1]*

**1** Department of Pathogen Biology, Guangdong Provincial Key Laboratory of Tropical Disease Research, Institute of Tropical Medicine, School of Public Health, Southern Medical University, Guangzhou, China, **2** Program in Public Health, University of California, Irvine, Irvine, California, United States of America

☯ These authors contributed equally to this work.

* xgchen@smu.edu.cn

**Data Availability Statement:** All relevant data are within the manuscript and its Supporting Information files.

**Funding:** This study was supported by a combination of funding from the National Nature

## Abstract

### Background

*Aedes albopictus* is one of the most invasive species in the world as well as the important vector for mosquito-borne diseases such as dengue fever, chikungunya fever and zika virus disease. Chemical control of mosquitoes is an effective method to control mosquito-borne diseases, however, the wide and improper application of insecticides for vector control has led to serious resistance problems. At present, there have been many reports on the resistance to pyrethroid insecticides in vector mosquitoes including deltamethrin to *Aedes albopictus*. However, the fitness cost and vector competence of deltamethrin resistant *Aedes albopictus* remain unknown. To understand the impact of insecticide resistant mosquito is of great significance for the prevention and control mosquitoes and mosquito-borne diseases.

### Methodology/Principal findings

A laboratory resistant strain (Lab-R) of *Aedes albopictus* was established by deltamethrin insecticide selecting from the laboratory susceptible strain (Lab-S). The life table between the two strains were comparatively analyzed. The average development time of Lab-R and Lab-S in larvae was 9.7 days and 8.2 days ($P < 0.005$), and in pupae was 2.0 days and 1.8 days respectively ($P > 0.05$), indicating that deltamethrin resistance prolongs the larval development time of resistant mosquitoes. The average survival time of resistant adults was significantly shorter than that of susceptible adults, while the body weight of resistant female adults was significantly higher than that of the susceptible females. We also compared the vector competence for dengue virus type-2 (DENV-2) between the two strains via RT-qPCR. Considering the results of infection rate (IR) and virus load, there was no difference between the two strains during the early period of infection (4, 7, 10 day post infection (dpi)). However, in the later period of infection (14 dpi), IR and virus load in heads, salivary glands and ovaries of the resistant mosquitoes were significantly lower than those of the suscepti-ble strain (IR of heads, salivary glands and ovaries: $P < 0.05$; virus load in heads and

Science Foundation of China (81829004, 31830087), the National Key Research and Development Program of China (2020YFC1200100), the National Institutes of Health, USA (AI136850) and the Guangzhou Synergy Innovation Key Program for Health (201807010005, 201803040006) to X-G.C. The funders had no role in study design, data collection and analysis, decision to publish, or preparation of the manuscript.

**Competing interests:** The authors have declared that no competing interests exist.

salivary glands: $P < 0.05$; virus load in ovaries: $P < 0.001$). And then, fourteen days after the DENV-2-infectious blood meal, females of the susceptible and resistant strains were allow to bite 5-day-old suckling mice. Both stains of mosquito can transmit DENV-2 to mice, but the onset of viremia was later in the mice biting by resistant group as well as lower virus copies in serum and brains, suggesting that the horizontal transmission of the resistant strain is lower than the susceptible strain. Meanwhile, we also detected IR of egg pools of the two strains on 14 dpi and found that the resistant strain were less capable of vertical transmission than susceptible mosquitoes. In addition, the average survival time of the resistant females infected with DENV-2 was 16 days, which was the shortest among the four groups of female mosquitoes, suggesting that deltamethrin resistance would shorten the life span of female *Aedes albopictus* infected with DENV-2.

## Conclusions/Significance

As *Aedes albopictus* developing high resistance to deltamethrin, the resistance prolonged the growth and development of larvae, shorten the life span of adults, as well as reduced the vector competence of resistant *Aedes albopictus* for DENV-2. It can be concluded that the resistance to deltamethrin in *Aedes albopictus* is a double-edged sword, which not only endow the mosquito survive under the pressure of insecticide, but also increase the fitness cost and decrease its vector competence. However, *Aedes albopictus* resistant to deltamethrin can still complete the external incubation period and transmit dengue virus, which remains a potential vector for dengue virus transmission and becomes a threat to public health. Therefore, we should pay high attention for the problem of insecticide resistance so that to better prevent and control mosquito-borne diseases.

### Author summary

Worldwide invasion and expansion of *Aedes albopictus*, the main vector of dengue, chikungunya, and Zika viruses, has become a serious concern in global public health. With the large use of insecticides, especially the most commonly used pyrethroid insecticides, the emergence and development of resistance in *Aedes albopictus* present vector control challenges. However, it is not clear whether the resistance would affect the fitness cost and vector competence of *Aedes albopictus*. In this study, a laboratory resistant strain of *Aedes albopictus* was established by selecting the susceptible strain of *Aedes albopictus* with deltamethrin. Comparing the resistant strain with the susceptible strain, we found that deltamethrin resistance increased the fitness cost and reduced the vector competence of DENV-2 in *Aedes albopictus*. These latest findings shared the light for dengue disease prevention and vector control strategies.

## Introduction

The Asian tiger mosquito [*Aedes albopictus* (Skuse)] is an invasive species of public health importance [1], originated in southeast Asia and has spread to all continents except Antarctica in the past 30–40 years [2]. *Ae. albopictus* is an important vector that transmits at least 20 viral

pathogens including dengue, chikungunya, and Zika viruses [3], and its aggressive bite to humans making it a serious health concern [4].

Dengue fever is one of the most rapidly spreading mosquito-borne viral diseases in the world, with 3.9 billion people in 128 countries or regions at risk [5], making dengue fever a global public health problem. Currently, no effective dengue vaccine or treatment has been developed to control dengue fever. The most effective way to prevent dengue fever relies on targeting the mosquito vector by using chemical insecticides. Pyrethroid insecticides are the most commonly used insecticides for adult mosquito control [6], among which deltamethrin is the representative insecticide of pyrethroid [7]. Due to the large-scale use of deltamethrin, resistance in mosquito field populations has been reported all over the world [8–10], and has developed rapidly [11].

It has been reported that insecticide resistance may lead to a series of side effects in the life history traits and vector competence of vector population. In *Aedes aegypt*i, study has shown that the survival time, wing length, fecundity of temephos resistant population were lower than those of susceptible population [12]. In *Anopheles gambiae*, Alout et al. revealed that the fecundity rate was reduced in insecticide resistant strains [13]. These studies indicated that the insecticide resistance may induce the fitness cost in mosquitoes. As for the vector competence of resistant mosquitoes, previous studies have showed that deltamethrin resistant *Anopheles gambiae* with *kdr* mutation is more susceptible to *Plasmodium falciparum* [14–16]. But Paeporn et al. reported that low level of temephos resistance in *Aedes aegypti* might not affect the susceptibility to dengue virus 2 [17]. Therefore, insecticides resistance may have an impact on the susceptibility of mosquitoes to pathogens, which poses a challenge to the prevention of mosquito-borne viral diseases. However, there are few reports about the impact of deltamethrin resistance on *Aedes albopicuts*.

In this study, we established a laboratory strain of *Aedes albopictus* resistant to deltamethrin, and compared the fitness cost and vector competence of DENV-2 between the resistant strain and the susceptible strain.

## Materials and methods

### Ethics statement

BALB/c suckling mice were purchased from the Animal Experimental Center of Southern Medical University, Guangdong Province, China. All animal experiments were conducted according to the guidelines established by the Association for Assessment and Acceleration of Laboratory Animal Care International. Animal experimental procedures were approved the Office of Laboratory Animal Welfare (approval number: A5867-01), and animal care was performed in accordance with institutional guidelines.

### Set-up of deltamethrin resistant strain

The laboratory susceptible strain (Lab-S), a strain of *Aedes albopictus* susceptible to deltamethrin provided by Shanghai Center for Disease Control and Prevention, has been kept in a laboratory environment for a long time without exposure to any insecticide. The laboratory resistant strain (Lab-R) is selected with deltamethrin for 27 generations from susceptible *Aedes albopictus* populations. Selection was performed by exposing each generation of fourth-stage larvae to a 50% lethal concentration ($LC_{50}$) of deltamethrin for 24 h. The $LC_{50}$ was determined by larval bioassay following WHO guidelines [18]. After 24 h, the surviving larvae were transferred to dechlorinated tap water for further feeding and generation. All the mosquitoes were maintained under standard insectary conditions (constant $27 \pm 1^\circ$C, 70–80% relative humidity, and a light: dark cycle of 14 h:10 h). The larvae (150–200 / L dechlorinated tap water water)

were reared and fed with turtle food. Adult mosquitoes were kept in cages (20 cm × 20 cm × 35 cm) and fed with a 10% glucose solution. The adult resistance bioassays were performed using deltamethrin insecticide (0.05%) following the standard WHO tube test protocol [19]. Testing kits and insecticide-impregnated papers with standard diagnostic doses were provided by the Universiti Sains Malaysia, Penang, Malaysia.

### Dengue virus

Dengue virus 2 (New Guinea C, GenBank: AF038403.1) was provided by the Key Laboratory of Tropical Disease Control of Sun Yat-sen University (Guangzhou, China). C6/36 cells were cultured in RPMI-1640 medium supplemented with 10% heat inactivated fetal bovine serum (FBS) and maintained at 28°C. Cells grown in a 25-cm$^2$ culture flask were inoculated with DENV-2 at 37°C for 2 days until obvious cytopathic effects. The supernatant was harvested and then the viruses were ready to be quantified with RT-qPCR before being blood-fed to the mosquitoes.

### Fitness cost analysis

**Life table at larvae stage.**   The Lab-S and Lab-R strains were simultaneously reared under identical conditions, such as initial larval density and feeding, temperature and illumination regimens. Eggs were induced to hatch for approximately 24 hours. Three replicates of 50 newly emerged larvae were then randomly transferred to plastic trays (9.5 × 6.7 × 6.2 cm) with 150 mL dechlorinated water and quantitative turtle food. New food supplement was offered every day. The kinetics of pupae formation and eclosion under the above conditions was accompanied daily as indicative of larval development time and pupal development time respectively.

**Adult mosquito survival.**   For the Lab-S and Lab-R strains, newly emerged (24h - old) adults resulting from the item above, were transferred to cages covered with nylon netting. After 2–3 days of mating, males were placed into paper bowls covered with gauze (40 mosquitoes / bowl). Separately the females of the Lab-S and Lab-R strains were divided into two parts. One was fed with blood meal containing cell culture supernatant as the control groups, and the other was infected with blood meal containing DENV-2 as the infection groups (DENV-S, DENV-R). After anesthesia with freezing, fully engorged mosquitoes were placed into paper bowls (35 mosquitoes / cup). 10% glucose solution was supplied to the mosquitoes and dead mosquitoes were recorded daily until all were dead.

**Adult mosquito fecundity.**   In the mosquitoes treated above, around 50 females from each group (control: Lab-S; DENV-S; control: Lab-R; DENV-R) were individualized in 250mL paper cups. The number of egglaying females and the amount of eggs / female in the second oviposition cycle were recorded.

**Measurement of adult mosquito size.**   The body size of adult mosquitoes was represented by wing length and body weight. Remove the wings of one mosquito and place it under a laboratory microscope (connected to a computer and a camera) to take photos. Image-pro Plus software was used to measure the wing length of adult mosquitoes. The wing length determined as the distance from the axillary incision to the apical margin (excluding fringes). The average length of a pair of wings was the mosquito's wing length. Mosquitoes were placed in an oven to dry for one hour, and their weight was measured in groups of 10 mosquitoes.

### Vector competence analysis

**Oral infections of mosquitoes.**   Mosquito infection was conducted in a Biological Safety Level 2 lab. The DENV-2 supernatant was collected and mixed with defibrinated sheep blood

at a ratio of 2: 1. The fresh DENV-2 was $10^5$–$10^7$ RNA copies/µL at each experiment. The blood meal was maintained at 37˚C for 30 min and transferred into a Hemotek blood reservoir unit (Discovery Workshops, L0061ncashire, United Kingdom). Five to seven-day-old female mosquitoes of two strain were starved for 20–24 h and allowed to feed on the infectious blood meal for 30 min. After anesthesia with freezing, fully engorged mosquitoes were removed and placed into 250-mL paper cups covered with gauze (10 mosquitoes / cup). All treatments were maintained at 80% relative humidity and 16 h: 8 h (light: dark) photoperiod, and all mosquitoes were fed 10% glucose solution on cotton pads placed on the net surface of each cup.

**Vector susceptibility analysis.** The midgut, head, ovaries, and salivary glands of each mosquito from the Lab-S strain and Lab-R strain were dissected and detected at 4, 7, 10 and 14 dpi. The sample size collected from each strain was 10 mosquitoes at each time point. The experiment was independently repeated three times. The legs and wings of each mosquito were removed, and disposable insect microneedles were used to separate the head, salivary glands, midgut and ovaries of each mosquito under an anatomical lens. Tissues were washed three times in PBS droplets and then transferred to 50 µL of TRIzol (Ambion, Life Technologies, Carlsbad, CA, United States) in 1.5-mL Eppendorf tubes. Total RNA was extracted according to TRIzol manufacturer's protocol. cDNA was synthesized using a random primer, and the recommendations of the GoScript Reverse Transcription System (Promega, Madison, WI, United States) were followed.

**Horizontal transmission assays.** Fourteen days after the DENV-2-infectious blood meal, the susceptible and resistant strain females were divided into 6 groups and starved for 10–12 h in advance. Each group was allow to bite 5-day-old suckling mice (fifteen mosquitoes / mouse) and feed for 1 h. Subsequently, all fully engorged mosquitoes were dissected to detect the infection rate of salivary glands and the virus load in the infected tissues by RT-qPCR. Blood were removed from the tail of individual mouse at 1, 3, 5 and 7 day post mosquitoes feeding and were detected by RT-qPCR. All the suckling mice were euthanized at 7 day post mosquitoes feeding and the brain of each individual mouse was dissected to detect the virus titer by RT-qPCR and a plaque assay [20].

**Vertical transmission assays.** Three days after the DENV 2-infectious blood meal, the susceptible and resistant females were allow to lay eggs for one week (first ovipositon cycle). Then, they were refed with defibrinated sheep blood to stimulate oviposition. Individual engorged females were placed into paper cups (250 mL) covered with gauze, containing a funnel-shaped filter papers. They were allow to lay eggs at 14dpi (second oviposition cycle). After one week, these eggs were counted and collected in a new 1.5-ml tubes. The ovaries of the mosquitoes were dissected for detecting DENV-2. Only the eggs coming from mosquitoes with DENV-2-positive ovaries who laid eggs were studied.

**Detection of dengue virus 2.** Absolute quantitative real-time PCR (RT-qPCR) was used to detect DENV-2 in the tissues of mosquitoes and infected mice. The plasmid standard was constructed as previously described [21]. The concentration of the plasmid was 1.36 ng/µL, which was transformed into a copy number ($4.41 \times 10^8$/µL). The RT-qPCR reaction mixture per well contained 10 mL SYBR selected master mix, 1 mL of each primer (10 mM), 2 mL cDNA or the plasmid standard, and 6 mL RNase-free water. The reaction was performed in the 7500 Real-Time PCR System as follows: 50˚C for 2 min, 95˚C for 2 min, followed by 40 cycles of 95˚C for 15 s, 60˚C for 15 s, and 72˚C for 1 min. Melting curves were given at 95˚C for 15 s, 60˚C for 1 min, 95˚C for 30 s, and 60˚C for 15 s. A standard curve was established by 10-fold dilutions of the plasmid standard ($4.41 \times 10^3$–$4.41 \times 10^7$). Each sample was conducted in three replicates, and the results were determined by the melting curve and cycle threshold values.

## Plague assay

BHK-21 cells ($2 \times 10^5$ cells / well) were plated in 24-well plates and incubated in a cell culture incubator until 90% to 95% confluency was reached. After centrifugation, the supernatant of the infectious brain was inoculated into the wells individually with serial dilutions. Methyl cellulose was used to overlay the infected cell monolayers. Five days after incubation, the cells were fixed with 4% paraformaldehyde solution for 1 h and then stained with crystal violet solution for 20 min at room temperature. The wells were washed with tap water and dried. The dilution that produced 10 to 50 plaques was selected, and the number of plaques for each replicate was counted. The plaque forming units per ml (PFU / ml) was calculated as follows: PFU / ml = average number of plaques / [(dilution factor of well) (volume of inoculum per plate)].

## Statistical analysis

All statistical analyses were performed with SPSS 20.0 (IBM). $LC_{50}$ and $KDT_{50}$ were estimated using the log-probit models. For larvae bioassays, the resistant status was measured by the resistant ratio ($RR_{50}$), i.e., the ratio of $LC_{50}$ for the Lab-R strain over $LC_{50}$ for the Lab-S strain. Larval resistance status was defined as susceptible if $RR_{50} < 5$, moderately resistant if $5 < RR_{50} < 10$, and highly resistant if $RR_{50} > 10$. For adult bioassays, resistant status was defined by mortality rate: Resistant if mortality < 90%, probably resistant if mortality was between 90 and 98%, and susceptible if mortality > 98%.

t-test was used to determine the differences between the Lab-S strain and Lab-R strain of *Aedes albopictus* in average development time from larvae to pupation, average development time from pupae to adult emergence, female fecundity, wing length and body weight. The difference in the survival time of adult mosquitoes was determined by Kaplan-Meier survival analysis and log-Rank test, and the difference in the median survival time was determined by Mann-Whitney U test.

Chi-square test was applied to compare the infection rate (IR) of tissues from susceptible and resistant strain of *Aedes albopictus* at the same time point (4 dpi, 7 dpi, 10 dpi and 14 dpi) and the difference of pupae formation rate, eclosion rate. The RNA copy number of DENV-2 was first logarithmically converted, and then t-test was used to compare the virus titer of DENV-2 in the tissues between susceptible and resistant strain at the same time point. $P < 0.05$ was considered statistically significant.

# Results

## Higher resistance to deltamethrin generated in Lab-R

Larva bioassay results showed that after 27 generations selected by deltamethrin, the $LC_{50}$ of deltamethrin increased from 0.0020 mg / L to 0.0435 mg / L in Lab-R strain larvae (Table 1). Compared with the Lab-S strain, $RR_{50}$ increased to 21.75 ($RR_{50} > 10$), reaching the high resistance standard. The mortality rate of adult mosquitoes of Lab-R strain decreased from 100% to 79.8%, which also met the criteria of resistant population (mortality < 90%). The Lab-R strain

**Table 1. Resistance bioassay of the laboratory—resistant strain.**

| Population | Adult bioassay | | Larval bioassay | |
|---|---|---|---|---|
| | KDT$_{50}$ (95% CI) (min) | CM (95% CI) (%) | LC$_{50}$ (95% CI) (mg / L) | RR$_{50}$ |
| Lab-S | 25 (21, 28) | 100.0 (100.0, 100.0) | 0.0020 (0.0010, 0.0010) | 1.0 |
| Lab-R | 32 (28, 35) | 79.8 (72.1, 88.0) | 0.0435 (0.0382, 0.0483) | 21.75 |

of *Aedes albopictus* could be identified as the resistant population by combining the two resistance indexes.

## Developing time in Lab-R much longer than Lab-S

By comparing the larval life table of the Lab-S strain and the Lab-R strain, it was found that the development time of the resistant strain from larvae to adult mosquitoes (11.5 days) were significantly longer than that of the susceptible strain (10.2 days) (t = 6.827, *df* = 4, *P* < 0.005). The development trend of Lab-R and Lab-S as shown in Fig 1A and 1B, the average larval development time of the resistant and susceptible strains was 9.7 days and 8.2 days respectively (t = 10.14, *df* = 4, *P* < 0.005), and the average pupal development time was 2.0 days and 1.8 days respectively (t = 2.177, *df* = 4, *P* > 0.05). The pupae formation rate of Lab-S strain (90.39%) and Lab-R strain (91.41%) were no significant difference (F = 0.086, *df* = 1, *P* > 0.05), and the eclosion rate of the Lab-S strain and Lab-R strain were 95.59% and 89.86%, with no significant difference (Fig 1B) (F = 3.084, *df* = 1, *P* > 0.05). These results indicate that deltamethrin resistance may affect the development of *Aedes albopictus* by prolong the time form larvae to develop into pupae.

## Survival time in Lab-R much shorter than Lab-S

Based on survival curve analysis, the average survival time of female mosquitoes in the Lab-S and Lab-R strains were 41 days and 23 days respectively, and the median survival time were 44 days and 24 days, respectively ($\chi^2$ = 47.688, *df* = 1, *P* < 0.001; Mann-Whitney U test *P* < 0.001). The survival curves of female mosquitoes from the Lab-S and Lab-R strains in the non—infection group (i.e., control group) were also significantly different (Fig 1C). The mean survival time in male mosquitoes of Lab-S strain and Lab-R strain was 26 days and 17 days respectively, and the median survival time was 26 days and 16 days respectively (Fig 1D) ($\chi^2$ = 41.210, *df* = 1, *P* < 0.001; Mann-Whitney U test *P* < 0.001).

After DENV-2 infection, the average survival time of female mosquitoes from the Lab-S strain in the infection group was 38 days, and the median survival time was 41 days, which was

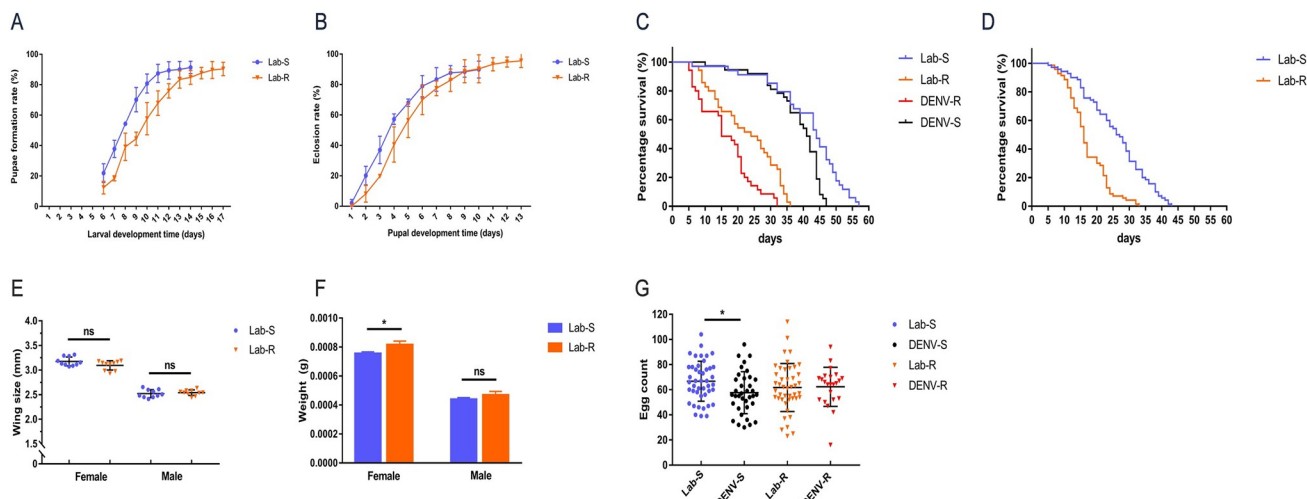

**Fig 1. Life table analysis of the susceptible and resistant strains of *Aedes albopictus*.** The results are represented as the mean ± standard error (SE). Blue represents the susceptible strain (Lab-S), and orange represents the resistant strain (Lab-R). (A) Larval developmental time. (B) Pupal developmental time. (C) Survival time of adult female *Aedes albopictus*. (D) Survival time of adult male *Aedes albopictus*. (E) Wing length of *Aedes albopictus*. (F) Weight of *Aedes albopictus*. (G) Fecundity of *Aedes albopictus*. * *P* < 0.05; ** *P* < 0.01; *** *P* < 0.005; ns:no significance.

significant shorter than that of control group (Fig 1C) ($\chi^2$ = 10.271, $df$ = 1, $P$ < 0.01; Mann-Whitney U test $P$ < 0.05). The average survival time of female mosquitoes in the infection group of Lab-R strain was 16 days, and the median survival time was 15 days, which was significantly shorter than that of control group (Fig 1C) ($\chi^2$ = 10.836, $df$ = 1, $P$ < 0.001; Mann-Whitney U test $P$ < 0.01). The average survival time and the median survival time of female mosquitoes in the infection group of Lab-R strain was significantly shorter than that of the infected Lab-S strain (Fig 1C) ($\chi^2$ = 65.879, $df$ = 1, $P$ < 0.001; Mann-Whitney U test $P$ < 0.001). These results displayed that deltamethrin resistance may affect the life span of adult *Aedes albopictus*, especially on the female mosquitoes infected with DENV-2.

## Significant difference in female weight but none in wing length between Lab-R and Lab-S strains

Both larvae from Lab-R strain and Lab-S strain were fed with the same amount of food and reared in the same space. There was no significant difference in wing length between female mosquitoes and between male mosquitoes of the two strains (Fig 1E) ($P$ > 0.05). The weight of resistant female mosquitoes was significantly higher than that of susceptible female mosquitoes ($P$ < 0.05), but no significant difference in weight between male mosquitoes of the two strains (Fig 1F) ($P$ > 0.05), indicating that deltamethrin resistance would affect the body size (weight) of female adult of *Aedes albopictus*.

## No significant difference in fecundity between Lab-R and Lab-S strains

After feeding on blood meal in the second ovipostion cycle, all female mosquitoes were allow to lay eggs in a single cup (Fig 1G). The average eggs number of one female mosquito in Lab-S group and Lab-R group were 67 and 61, respectively, with no statistical difference ($P$ > 0.05). The average egg number of one female mosquito in the DENV-S group was 56, which was significantly less than that in the Lab-S group (t = 2.489, $df$ = 78, $P$ < 0.05). While the average egg number of one female mosquito in the DENV-R group was 64, with no significant difference compared with the Lab-R group, implying that deltamethrin resistance would not affect the fecundity of female adult of *Aedes albopictus*.

## Susceptibility to DENV-2 significantly lower in Lab-R than Lab-S in the later period of infection

To evaluate the susceptibility to DENV-2 in the Lab-R and Lab-S strains, 30 mosquitoes of each strain were used to detect DENV-2 infection by RT-qPCR at 4, 7, 10 and 14 dpi. The IR of tissues in resistant and susceptible strains of *Aedes albopictus* showed different trends with the time of days post-infection. With the days of infection, from 4dpi to 14dpi, susceptible strain of *Aedes albopictus* broke through the midgut barrier and spread to the heads, salivary glands and ovaries, and IR of tissues gradually increased. However, the IR in the tissues of Lab-R strain decreased at 14dpi after increasing at 10dpi (Fig 2A–2D). At the four infection time points, the IR in the midguts of the two strains were between 70% and 90% (Fig 2A), with no significant difference. At the first three infection time points (4 dpi, 7 dpi, 10 dpi), there was no difference in the IR between the two strains, but at 14 dpi, the IR of heads, salivary glands and ovaries of the Lab-R strain were significantly lower than that of the Lab-S strain (Fig 2B–2D) ($P$ < 0.05).

The DENV-2 RNA copies in each tissue of the Lab-R strain and Lab-S strain were 6–8 copies (log10)/μL at each infection time point. At 7 dpi, the number of DENV-2 RNA copies in the midguts of the Lab-R strains was significantly lower than that of the Lab-S strains (Fig 2E)

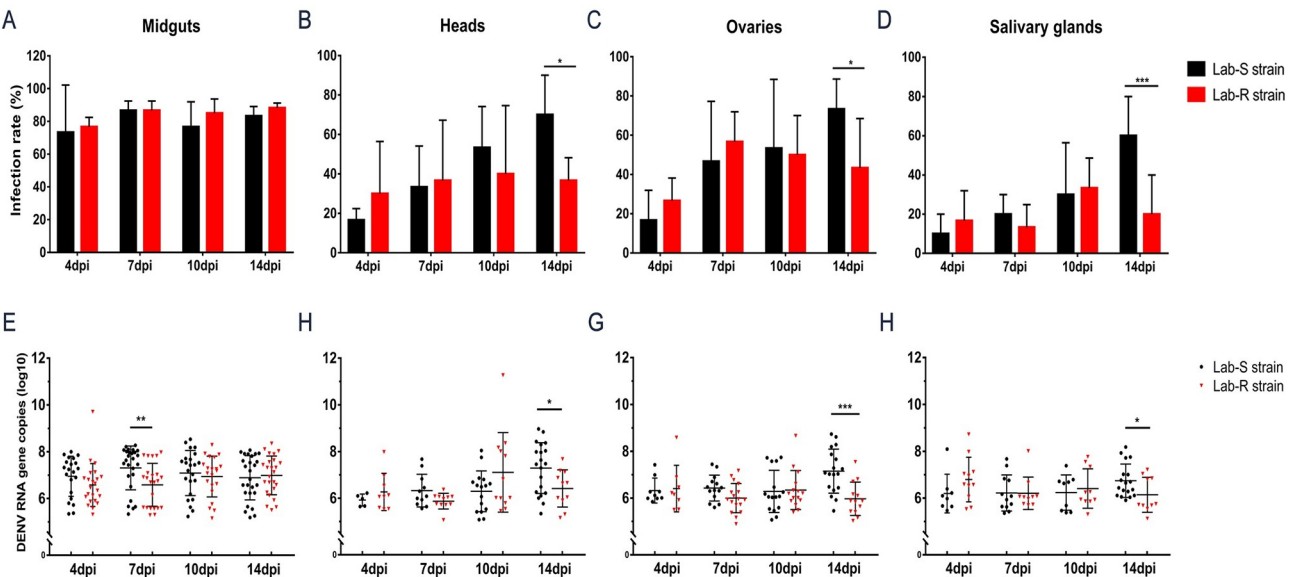

**Fig 2. Spatio-Temporal infections to DENV-2 in susceptible and resistant strains of *Aedes albopictus*.** The midguts, heads, ovaries and salivary glands from the two strain were dissected at 4, 7, 10, and 14 day post-infection, and DENV-2 virus was detected by RT-qPCR. The experiment was repeated three time. Lab-S strain, n = 30 at each time point; Lab-R strain, n = 30 at each time point. The results are expressed as the mean ± standard error (SE). (A-D) Infection rate of midguts, heads, ovaries, salivary glands. (E-H) DENV-2 RNA copies in infected midguts, heads, ovaries, salivary glands. $^*$ $P < 0.05$; $^{**}$ $P < 0.01$; $^{***}$ $P < 0.005$.

($P < 0.01$). At 14dpi, the number of DENV-2 RNA copies in the heads and salivary glands of the Lab-R strain was also significantly lower than that of the Lab-S strain (Fig 2F and 2G) ($P < 0.05$). Furthermore, the virus load in the ovaries of the Lab-R strain was also significantly lower than that of the Lab-S strain (Fig 2H) ($P < 0.001$), which indicated that high deltamethrin resistance is likely to reduce the susceptibility to DENV-2 in *Aedes albopictus* in the later period of infection.

## Horizontal transmission of DENV-2 between Lab-R and Lab-S strains

To evaluate the horizontal transmission of DENV-2 of two strains in the later infection period, suckling mice were bitten by 15 females fed on DENV-2-infectious blood meal and the infection of mice was used as the basis of evaluation. In each group, the number of virus positive salivary glands of blood-engorged females ranged from 2 to 7 and the mean virus copy number (log10) / μL of each group ranged from 5.68 to 7.99 (Table 2). The DENV-2 was detected in all six suckling mice of the Lab-S group, and one of the Lab-R group was not infected. The DENV-2 viremia was earlier detected in the susceptible group than the resistant group. Day one after the bite, two mice in the susceptible group could be detected the virus, whereas the viremia was only observed on the day 3 after the bite in the resistant group. During 1–7 days after biting, the virus copy number (log10) / μL in serum of susceptible group ranged from 2.66 to 3.87 was higher than that in the resistant group (2.05 to 3.42 (log10) / μL). Furthermore, DENV-2 were detected by RT-qPCR in all of mouse brains which displayed viremia and the average copies of DENV-2 in brains of susceptible group (5.30 (log10) / μL), with the maximum copies number of virus (12.20 (log10) / μL), was higher than that in the resistant group (4.33 (log10) / μL). But only the mouse brain with the highest virus copies can be also detected by the plaque assay, and it was $4 \times 10^5$ PFU / mL. These positive results demonstrate that there was no significant correlation between the suckling mice infected and the number of salivary

**Table 2. Transmission of DENV-2 to suckling mice by infected *Aedes albopictus* bite.**

| Group | No. salivary gland positive mosquitoes (mean copies log10 / μL) | DENV-2 copies in serum (log10) / μL | | | | DENV-2 copies in brain (log10) / μL (virus titer†) |
|---|---|---|---|---|---|---|
| | | Day 1 | Day 3 | Day 5 | Day 7 | Day 7 |
| **Lab-S** | | | | | | |
| S1 | 2 (5.73) | 2.66 | − | 2.82 | − | 3.32 |
| S2 | 3 (5.68) | − | 3.42 | − | − | 3.72 |
| S3 | 3 (6.33) | − | − | − | 2.93 | 3.12 |
| S4 | 3 (7.26) | 3.12 | 2.98 | − | 2.87 | 12.20 ($4 \times 10^5$) |
| S5 | 4 (5.76) | − | − | 2.89 | 3.87 | 4.22 |
| S6 | 6 (6.51) | − | − | − | 3.73 | 5.20 |
| **Lab-R** | | | | | | |
| R1 | 2 (7.99) | − | − | 3.42 | 1.79 | 2.63 |
| R2 | 4 (6.45) | − | − | − | 3.08 | 3.46 |
| R3 | 5 (6.55) | − | − | 3.01 | 2.70 | 4.01 |
| R4 | 5 (6.66) | − | 2.71 | 2.90 | 2.05 | 8.31 |
| R5 | 6 (7.56) | − | − | − | − | − |
| R6 | 7 (7.14) | − | − | − | 2.87 | 3.26 |

† Vitus titer in brain (PFU / ml)

−: negative

gland positive mosquitoes or the virus copy number in salivary glands. However, the later onset of viremia and the lower viral titer in serum and brain tissues of the mice bitten by resistant mosquitoes indicated that the horizontal transmission of DENV-2 by resistant strain was lower than that of the susceptible strain.

## Vertical transmission of DENV-2 between Lab-R and Lab-S strains

To assess the vertical transmission of DENV-2 of the Lab-R and Lab-S strains, infective rate of egg pools and minimal filial infection rate (MFIR) were calculated. As Table 3 shows, a total of 52 and 47 female mosquitoes respectively in the susceptible and resistant strains had oviposition success in the second ovipositon cycle. The infection rate in the ovaries of susceptible females (68.76%) was significantly higher than that of resistant females (48.61%) ($P < 0.05$), which was the same as the experimental results above. The eggs laid by a single female mosquito were detected in a pool, and the eggs infection rate of Lab-S strain (66.94%) was significant higher than that of Lab-R strain (35.00%) ($P < 0.05$), which suggested that the horizontal transmission ability of the resistant strain was lower. But there was no significant difference between the minimal filial infection rate of Lab-S strain and Lab-R strain, which were 1.16% and 0.695% respectively ($P > 0.05$).

**Table 3. Vertical transmission of DENV-2 by infected *Aedes albopictus* populations.**

| Population | No. tested females | No. ovary positive females (Infection rate) | Total egg number | No. egg pools with DENV-2-positive | Infection rate of egg pools (%)* | Minimal filial infection rate (MFIR, %) |
|---|---|---|---|---|---|---|
| **Lab-S** | 52 | 36 (68.76%) | 2075 | 24 | 66.94† | 1.16‡ |
| **Lab-R** | 47 | 23 (48.61%) | 1433 | 8 | 35.00† | 0.695‡ |

*: Statistical test, $P < 0.05$

† Infection rate of egg pools = the number of egg pools with DENV– 2-positive / the number of ovary positive females.

‡ Minimal filial infection rate (MFIR) = the number of egg pools with DENV-2-positive / total egg number.

## Discussion

In this study, we successfully established a laboratory deltamethrin resistant strain of *Aedes albopictus*. The Lab-S strain and the Lab-R strain shared a common genetic background, and differed only in the susceptibility to deltamethrin insecticide. It is the first time that we have experimentally demonstrated that deltamethrin resistance significantly increased the fitness cost in *Aedes albopictus* and decreased its vector competence to DENV-2.

Vector longevity is an essential factor in disease transmission. The good vitality of mosquitoes can increase the potential for infective bites to hosts and increase the possibility of infection. Life span is particularly important for viruses with an external incubation period, which allows the virus to complete the process of development in the vector [22]. However, the emergence of insecticides resistance is often accompanied by the fitness cost in mosquitoes, such as the shortening of life span and the decrease of fecundity. In this study, we found that deltamethrin resistance prolonged the development time and shorten the life span of *Aedes albopictus*, which was consistent with the previous studies on the slower development time of *Aedes aegypti* resistant to pyrethroids [23]. Moreover, organophosphorus resistance in *Culex pipiens* has been associated with a lower longevity in the laboratory [24] and in the field during winter, suggesting some resistance genes such as *Ace* gene may have impact on adult survival [25, 26]. Besides, the fecundity of resistant strain did not change significantly, while previous studies reported that the fecundity of highly pyrethroids resistant *Aedes aegypt* decreased [23]. In addition, our results showed that the body weight of resistant females was significantly higher than that of susceptible females, but there was no difference in wing length. Similarly, in *Culex pipiens*, there was no significant difference in wing length between the resistant and susceptible strains [24]. In addition to some resistance genes, there are two main mechanisms for the reduction of mosquito longevity caused by insecticides resistance: resource-based trade-offs and oxidative stress. For insects, when resources are limited, an increased investment in certain fitness-related traits tends to be accompanied by a significant reduction in lifespan. Insecticide resistance will induce the overexpression of some detoxifying enzymes in vector, which consumes a lot of resources in the body. A large amount of proteins are used to synthesize detoxifying enzymes, which may reduce the survival fuel like lipids in insects [27]. Oxidative stress is caused by a mismatch between the production of damaging reactive oxygen species (ROS) and that of protective antioxidants. The production of excess ROS has irreversible and harmful effects on the body and is widely regarded as a mechanism of aging. Overexpression of p450 monooxidase and GST, which are related to insecticides resistance, may greatly break the balance of ROS and produce excessive harmful ROS in insects [28, 29], resulting in the fitness cost.

In this study, high deltamethrin resistance and vector fitness cost are likely to decrease the dissemination and transmission of DENV-2 in *Aedes albopictus* in the later period of infection, which differs from previous studies. It has been reported that resistant *Anopheles* mosquitoes with *kdr* mutation increased the susceptibility to *plasmodium infection* [15], and the vector competence of *Culex quinquefasciatus* resistant to organophosphate for West Nile virus was also increased while its vector competence for Rift Valley Fever virus did not change [30], suggesting different interactions between different pathogens and vectors. Subsequently, we assessed the horizontal transmission between the two strains in the later period of infection (14 dpi). A related research conducted by an animal model found that a higher number of infectious mosquitoes delivered a higher total dose of zika virus, and drove earlier onset of viremia, higher magnitude of viremia [31]. Our study showed that the DENV-2 was detected later in the serum of suckling mice bitten by the resistant strain, suggesting that the horizontal transmission ability of the resistant strain was lower. It is possible that resistant mosquitoes

may bite suckling mice less than susceptible mosquitoes, and that each infected mosquito may transmit lower total dose virus to the mice. Using appropriate animal models to evaluate the vector horizontal transmission to hosts should be a feasible method to simulate the transmission of mosquito-borne diseases in the field. At the same time, we also found that the vertical transmission ability of the resistant strain was lower than that of the susceptible strain. Based on the above results, it can be concluded that the susceptibility, horizontal transmission and vertical transmission of resistant strain for DENV-2 were lower than those of susceptible strain.

Previous studies have suggested that life span and fecundity of *Aedes aegypti* are reduced post infection with dengue virus [32, 33]. Similarly, in this study, the life span of female *Aedes albopictus* was reduced post-infection, and the fecundity of susceptible strain was reduced after infection with dengue virus, while the fecundity of resistant strain did not change. Meanwhile, the lifespan of susceptible strain was reduced less than that of resistant strain after infection, which may indicate that deltamethrin resistance will further shorten the lifespan of female *Aedes albopictus* infected with dengue virus and increase the fitness cost of resistant mosquitoes.

Under selected pressure from insecticides, mosquitoes make a choice for survival, and the lack of resources available to fight against viral infections decrease their vector competence. The emergence of resistance is often associated to the overexpression of metabolic detoxification enzymes, which will deplete the energy reserve in mosquitoes, thus reducing the resources that can be used for other biological functions (lipids, glycogen, etc.), and may also produce excessive oxidative stress reaction to affect the immune response in mosquitoes [34]. Moreover, the signal mediated by the redox-sensing gene *Nrf2* is a major pleiotropic regulator for midgut homeostasis. In *Aedes aegypti*, the mosquitoes with *Nrf2*-silencing had lower microbiota level and the reduction of zika virus infection, and this gene is involved with insecticide resistance [35]. Therefore, the homeostasis of oxidative stress in resistant strain and susceptible strain may be different, leading to differences in longevity and vector competence. Some studies also found that the abundance of microbiota in resistant strain of *Anopheles* mosquitoes was lower than that in susceptible strain, which may be one of the factors affecting the vector competence between the two strains [36]. In addition, mutations in resistance gene loci may affect the expression of related immune genes in *Anopheles* mosquitoes through tight linkage of resistance gene with other gene(s) on the haplotype [16], but more experimental evidence is still needed for further study.

In conclusion, resistance to deltamethrin significantly increased fitness cost of *Aedes albopictus*, and decreased its vector competence for DENV-2. Mosquitoes can survive under selecting pressure of insecticides by developing resistance in order to restore vector population density, making the rapid development of insecticides resistance have become a problem for vector control. Although the vector competence of resistant *Aedes albopictus* is reduced, it can still complete the external incubation period and transmit dengue virus, which is a key point in vector control strategy that cannot be ignored. Therefore, the rational use of insecticides and the management of resistance are still urgent problems to be solved in the prevention and control of mosquitoes and mosquito-borne viral diseases.

## Acknowledgments

The authors would like to thank Dr. Hongxia Chen from Shanghai Center for Disease Control and Prevention for providing the susceptible strain of *Aedes albopictus* for our study, and colleagues from Southern Medical University who provided advice, helped with experiments and provided valuable support.

## Author Contributions

**Conceptualization:** Jielin Deng, Yijia Guo, Xiao-Guang Chen.

**Data curation:** Jielin Deng, Yijia Guo, Yang Wu.

**Formal analysis:** Jielin Deng, Yijia Guo, Xinghua Su, Shuang Liu, Wenqiang Yang, Yang Wu.

**Funding acquisition:** Xiao-Guang Chen.

**Investigation:** Guiyun Yan.

**Methodology:** Jielin Deng, Yijia Guo, Xinghua Su, Shuang Liu, Wenqiang Yang, Yang Wu.

**Project administration:** Kun Wu, Xiao-Guang Chen.

**Resources:** Kun Wu.

**Software:** Jielin Deng, Yijia Guo, Xinghua Su.

**Supervision:** Kun Wu, Guiyun Yan, Xiao-Guang Chen.

**Validation:** Jielin Deng, Yijia Guo.

**Visualization:** Jielin Deng, Yijia Guo.

**Writing – original draft:** Jielin Deng, Yijia Guo, Xiao-Guang Chen.

**Writing – review & editing:** Xiao-Guang Chen.

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
