## [Decision Letter · Decision Letter 0]

26 Mar 2021

Dear Professor Chen,

Thank you very much for submitting your manuscript "Impact of deltamethrin-resistance in Aedes albopictus on its fitness cost and vector competence" for consideration at PLOS Neglected Tropical Diseases. As with all papers reviewed by the journal, your manuscript was reviewed by members of the editorial board and by several independent reviewers. The reviewers appreciated the attention to an important topic. Based on the reviews, we are likely to accept this manuscript for publication, providing that you modify the manuscript according to the review recommendations. 

Sincerely,

Thomas S. Churcher

Associate Editor

Tereza Magalhaes

Deputy Editor

Reviewer's Responses to Questions

**Key Review Criteria Required for Acceptance?**

**Methods**

-Are the objectives of the study clearly articulated with a clear testable hypothesis stated?

-Is the study design appropriate to address the stated objectives?

-Is the population clearly described and appropriate for the hypothesis being tested?

-Is the sample size sufficient to ensure adequate power to address the hypothesis being tested?

-Were correct statistical analysis used to support conclusions?

-Are there concerns about ethical or regulatory requirements being met?

Reviewer #1: (No Response)

Reviewer #2: Very clearly presented methodology in all aspects, descent hypothesis, well executed study

Reviewer #3: The study has a clear testable hypothesis stated，appropriate design and sufficient sample. There were not concerns about ethical or regulatory requirements.

**Results**

-Does the analysis presented match the analysis plan?

-Are the results clearly and completely presented?

-Are the figures (Tables, Images) of sufficient quality for clarity?

Reviewer #1: (No Response)

Reviewer #2: the analysis is comprehensive and the results clearly and completely presented. The quality of illustration is very good. It is a very well and high quality manuscript.

Reviewer #3: 1. Table1，add CM (%)（95% CI.

2.Line273, Which generation started to make a difference？the value of t OR F？What caused the difference? Is there a correlation Developing time and level of resistance？

3.Line280 The value of t OR F？

4.Table 2 SI why DENV-2 copies in serum were observed in DAY1 and DAY5, but negative in DAY3 and DAY7, Is this a detection problem or what? The same problem appeared in S4.

**Conclusions**

-Are the conclusions supported by the data presented?

-Are the limitations of analysis clearly described?

-Do the authors discuss how these data can be helpful to advance our understanding of the topic under study?

-Is public health relevance addressed?

Reviewer #1: (No Response)

Reviewer #2: The conclusions are correct and carefully drawn and the fact that the two strains have similar genetic background is in support. However, the (negative in this case) correlation between insecticide resistance and fitness cost/infectivity may not be a general correlation between phenotypes, but specific for the mechanism selected by the approach taken (extensive laboratory selection for many generation), in that case the conclusion might also be more generally applicable and valid. 

It would be very useful, with not much more effort I believe, if the authors sequence the pyrethroid target in the regions kdr type mutations have been reported in this species in the two strains, to find out if target site resistance or no target resistance is responsible for the phenotype, and thus open the applicability of their conclusion. If resistance has been characterized already in other study, then please just cite the study and mention the dominant mechanism.

Reviewer #3: Line35.36 Is there no difference between resistant and sensitive strains during the early period of infection? Is it generally believed that there is a difference? why?

The vertical and horizontal transmission of resistant strains is lower than that of susceptible strains. Why is the level of drug resistance increasing worldwide, but dengue outbreaks are more frequent?

**Editorial and Data Presentation Modifications?**

Reviewer #1: (No Response)

Reviewer #2: (No Response)

Reviewer #3: Accept

**Summary and General Comments**

Reviewer #1: (No Response)

Reviewer #2: It would be useful to please sequence the pyrethroid target in the region of kdr type mutations previously found in Ae albopictus, to define if target site resistance is responsible or not for the phenotype, and thus open the applicability of the correlation conclusions.

Reviewer #3: (No Response)

PLOS authors have the option to publish the peer review history of their article (what does this mean?). If published, this will include your full peer review and any attached files.

Reviewer #1: Yes: Magellan TCHOUAKUI

Reviewer #2: Yes: John Vontas

Reviewer #3: No

Figure Files:

Data Requirements:

Reproducibility:

References

---

## [Editor Report · Decision Letter 1]

15 Apr 2021

Dear Professor Chen,

We are pleased to inform you that your manuscript 'Impact of deltamethrin-resistance in Aedes albopictus on its fitness cost and vector competence' has been provisionally accepted for publication in PLOS Neglected Tropical Diseases.

Best regards,

Thomas S. Churcher

Associate Editor

Tereza Magalhaes

Deputy Editor

---

## [Editor Report · Acceptance letter]

22 Apr 2021

Dear Professor Chen,

We are delighted to inform you that your manuscript, "Impact of deltamethrin-resistance in Aedes albopictus on its fitness cost and vector competence," has been formally accepted for publication in PLOS Neglected Tropical Diseases.

Best regards,

Shaden Kamhawi

co-Editor-in-Chief

Paul Brindley

co-Editor-in-Chief
